

# Non-adiabatic linear response in open quantum systems

Xiaotian Nie[1] and Wei Zheng[1,2,3*]

**1** Hefei National Laboratory, Hefei 230088, China
**2** Hefei National Research Center for Physical Sciences
at the Microscale and School of Physical Sciences,
University of Science and Technology of China, Hefei 230026, China
**3** CAS Center for Excellence in Quantum Information and Quantum Physics,
University of Science and Technology of China, Hefei 230026, China

⋆ zw8796@ustc.edu.cn

## Abstract

Adiabatic theorem and non-adiabatic corrections have been widely applied in modern quantum technology. Recently, non-adiabatic linear response theory has been developed to probe the many-body correlations in closed systems. In this work, we generalize the non-adiabatic linear response theory to open quantum many-body systems. We show that, similar to the closed case, the first-order deviation from the instantaneous steady state is memoryless—it depends only on the final parameters and not on the initial state or ramping path. When ramping the Hamiltonian, the linear response of observables is governed by the derivative of the retarded Green's function, as in closed systems. In contrast, ramping the dissipation gives rise to a different response, characterized by a high-order correlation function defined in the steady state. Our results offer a compact and physically transparent formulation of non-adiabatic response in open systems, and demonstrate that ramping dynamics can serve as a versatile tool for probing many-body correlations beyond equilibrium.



# 1 Introduction

The adiabatic theorem is one of the most powerful tools in quantum mechanics. It states that if a Hamiltonian varies sufficiently slowly, the system remains in its instantaneous eigenstate, up to a path-dependent phase factor known as the Berry phase [1]. The adiabatic theorem has vast applications in modern quantum technologies. For instance, it enables preparation of the desired quantum states in quantum information processes. Moreover, adiabatic processes have been designed for the adiabatic quantum computation, which serves an alternative to the conventional quantum computing based on quantum circuits [2–4]. Furthermore, during adiabatic evolution, eigenstates accumulate extra Berry phases, which are deeply connected to a variety of geometric and topological phenomena, including the Aharonov-Bohm effect, quantum Hall effects, and the physics of topological materials [5–12].

Beyond the adiabatic limit, a finite ramping velocity can induce transitions to other instantaneous non-steady eigenstates, leading to non-adiabatic excitations. Such non-adiabatic corrections also inspire studies on interesting ramping dynamics such as Kibble-Zurek scaling crossing phase transitions and Thouless topological charge pumping [13–18]. More recently, linear response theory beyond adiabatic ramping has been proposed as a powerful tool to probe many-body correlations [19]. This approach has already been applied to detect critical behavior in experiments with bosonic atoms in optical lattices.

Recently, the concept of adiabaticity and linear response theory have been generalized from closed to open quantum systems [20–49]. The adiabatic approximation and its validity in open quantum systems were first proposed in [21]. Beyond the adiabatic limit, the Kibble-Zurek mechanism [43, 44, 50] and the Thouless topological pumping have been proposed in open systems [37, 39, 40, 42].

Among these developments, three existing works are particularly relevant to the formulation of non-adiabatic response theory in open quantum systems. First, Ref. [30] shows that in the adiabatic limit, the deviation from the non-equilibrium steady state scales linearly with the ramping velocity. However, the deviation is characterized only through matrix norms, without resolving how it manifests in measurable quantities such as observables or correlation functions. Second, Ref. [27] presents a mathematically rigorous recursive expansion for non-adiabatic corrections to all orders. However, the expressions are given in a formal integral form over the entire evolution path, making them difficult to apply in practice, especially in many-body systems. Moreover, the structure of the first-order correction is not isolated or interpreted physically. Third, Ref. [31] focuses mainly on the dynamics within a degenerate steady-state subspace, analyzing leakage due to non-adiabatic effects. While this work is insightful for degenerate cases, the non-adiabatic corrections are again formulated in terms of complicated integrals that are abstract and not easily connected to experimentally relevant observables.

In this work, we generalize the non-adiabatic linear response theory from closed to open quantum systems governed by Lindblad dynamics [51,52]. Focusing on systems with a unique steady state, we show that under slow parameter ramping, the system follows the instantaneous steady state up to a linear correction in the ramping velocity. Our key findings are twofold. First, we demonstrate that the first-order correction is memoryless—it depends only on the system's final properties and not on the initial state or the details of the ramping path. Second, and more importantly, we derive a concise, universal, and physically transparent expression for the deviation in observable expectation values:

$$\Delta O = \left\langle \hat{O}(t_{\mathrm{f}}) \right\rangle - \mathrm{Tr}\left( \hat{O} \hat{\rho}_{\mathrm{ss}}(t_{\mathrm{f}}) \right)$$
$$= \mathrm{i} \frac{\partial \mathcal{G}(\omega)}{\partial \omega}\bigg|_{\omega=0} \cdot v + \cdots, \tag{1}$$

where $\mathcal{G}$ is a many-body correlation function. While in closed systems the ramping typically involves only the Hamiltonian, open systems allow both the Hamiltonian and the dissipation operators to be ramped. We show that ramping the Hamiltonian yields the standard retarded Green's function $\mathcal{G} = G^{\mathrm{R}}$, recovering results known from closed systems [19]. In contrast, ramping the dissipation gives rise to a different response, governed by a higher-order steady-state correlation function $\mathcal{G} = \mathcal{C}^{\mathrm{R}}$. We validate our results in two many-body models: the dissipative PXP model and the dissipative Dicke model, and find excellent agreement between theory and numerics for both Hamiltonian and dissipative ramping protocols. Finally, we show in the Appendix A that exceptional points (EPs) — where the Lindbladian becomes non-diagonalizable — do not affect our conclusions. Our non-adiabatic linear response theory provides a practical and broadly applicable framework for probing many-body correlations in open systems and for interpreting ramping dynamics in quantum simulation and control platforms.

## 2 Non-adiabatic linear response theory

### 2.1 Vectorization mapping

Within a quantum system weakly coupled to a Markovian bath, its dynamics should be described by the Lindblad master equation,

$$\mathrm{i}\partial_t \hat{\rho} = \left[\hat{H}, \hat{\rho}\right] + \mathrm{i}\sum_i \kappa_i \left(2\hat{J}_i \hat{\rho} \hat{J}_i^\dagger - \left\{\hat{J}_i^\dagger \hat{J}_i, \hat{\rho}\right\}\right) \coloneqq \mathring{\mathcal{L}}[\hat{\rho}], \tag{2}$$

where $\hat{\rho}$ is the density matrix of the system, $\hat{H}$ is the Hamiltonian dominating the coherent dynamics, and $\hat{J}_i$ are the jump operators characterizing dissipation with rates $\kappa_i$. The right-hand side defines the Lindbladian superoperator $\mathring{\mathcal{L}}$. We note that, in our convention, the Lindbladian includes an imaginary unit factor to make it resemble the Schrödinger equation of closed systems. This choice of convention ensures that the eigenvalues of $\mathring{\mathcal{L}}$ lie in the lower half of the complex plane and are symmetric about the imaginary axis.

To facilitate analysis, it is convenient to represent the density matrix and the superoperator in vectorized form using the Choi-Jamiołkowski isomorphism (also known as vectorization) [51,53]. This mapping transforms superoperators acting on both sides of the density matrix into matrices acting linearly on a single vector in an enlarged space.

Given a complete set of bases of the $d$-dimensional Hilbert space $\{|i\rangle\}$, the vectorization maps an arbitrary operator to a state in the $d^2$-dimensional doubled space, $|i\rangle\langle j| \to |j\rangle \otimes |i\rangle$. Accordingly, a density matrix $\hat{\rho}$ is mapped to a state vector $|\rho\rangle$, and any superoperator acting

on both sides of the density matrix is now represented by a huge matrix acting on the state,

$$\hat{\rho} = \sum_{i,j} \rho_{i,j} |i\rangle \langle j| \rightarrow |\rho\rangle = \sum_{i,j} \rho_{i,j} |j\rangle \otimes |i\rangle \,, \tag{3}$$

$$\hat{A}\hat{\rho}\hat{B} \rightarrow \left(\hat{B}^{\mathrm{T}} \otimes \hat{A}\right)|\rho\rangle \,. \tag{4}$$

Thus, the Lindblad master equation transforms into a Schrödinger-like equation in the doubled space: $\mathrm{i}\partial_t |\rho\rangle = \hat{\mathcal{L}} |\rho\rangle$, where $\hat{\mathcal{L}}$ is given by

$$\hat{\mathcal{L}} \rightarrow \hat{\mathcal{L}} = \left(\hat{I} \otimes \hat{H} - \hat{H}^{\mathrm{T}} \otimes \hat{I}\right) + \mathrm{i} \sum_i \kappa_i \left[ 2\hat{J}_i^* \otimes \hat{J}_i - \hat{I} \otimes \hat{J}_i^\dagger \hat{J}_i - \left(\hat{J}_i^\dagger \hat{J}_i\right)^{\mathrm{T}} \otimes \hat{I} \right] \,. \tag{5}$$

And the Hilbert-Schmidt inner product of matrices is mapped to the state inner product,

$$\mathrm{Tr}\left(\hat{\rho}_m^\dagger \hat{\rho}_n\right) \rightarrow \langle \rho_m | \rho_n \rangle \,, \tag{6}$$

which is a useful relation while calculating expectation values. The inverse of the vectorization mapping is also direct: reshaping the column vector to a square matrix.

## 2.2 First-order correction of density matrix

Via the vectorization mapping, the Lindblad equation is transformed to a Schrödinger-like equation in the doubled space $\mathrm{i}\partial_t |\rho\rangle = \hat{\mathcal{L}} |\rho\rangle$, where the Lindbladian now serves as a non-Hermitian Hamiltonian dominating the evolution of state $|\rho\rangle$.

Let us consider a slow time-dependent change in the parameter $\lambda(t)$, varying from $\lambda(t_i) = \lambda_i$ to $\lambda(t_f) = \lambda_f$ over a finite time interval. $\lambda(t)$ may appear either in the Hamiltonian or in the dissipation, which will be specified later. The evolution of the system is then described by a time-dependent equation,

$$\mathrm{i}\partial_t |\rho(\lambda(t))\rangle = \hat{\mathcal{L}}(\lambda(t)) |\rho(\lambda(t))\rangle \,. \tag{7}$$

Generally speaking, a non-Hermitian matrix is not always guaranteed to be diagonalizable. In particular, exceptional points (EPs) — where two or more eigenvectors coalesce — can arise in the parameter space [54–56]. However, such points form a measure-zero subset in matrix space and can be avoided by infinitesimal perturbations of the parameter trajectory. Therefore, diagonalizability is the generic case for non-Hermitian matrices encountered in practice. Throughout this section, we assume that the Lindbladian is diagonalizable and non-degenerate during the ramping process. The behavior near exceptional points is analyzed separately in the Appendix A.

Unlike Hermitian systems, where left and right eigenvectors are Hermitian conjugates of each other, a non-Hermitian Lindbladian has distinct left and right eigenvectors, satisfying biorthogonality:

$$\hat{\mathcal{L}} \left|\rho_m^{\mathrm{R}}\right\rangle = E_m \left|\rho_m^{\mathrm{R}}\right\rangle \,, \qquad \left\langle \rho_m^{\mathrm{L}}\right| \hat{\mathcal{L}} = E_m \left\langle \rho_m^{\mathrm{L}}\right| \,, \tag{8}$$

$$\left\langle \rho_m^{\mathrm{L}} \middle| \rho_n^{\mathrm{R}} \right\rangle = \delta_{mn} \,, \qquad\qquad \hat{\mathcal{L}} = \sum_m E_m \left|\rho_m^{\mathrm{R}}\right\rangle \left\langle \rho_m^{\mathrm{L}}\right| \,, \tag{9}$$

where $E_m$ are complex eigenvalues with non-positive imaginary parts, $\left\langle \rho_m^{\mathrm{L}}\right|$ and $\left|\rho_m^{\mathrm{R}}\right\rangle$ are the corresponding left and right eigenvectors. Among them, there is always one null eigenvalue $E_0 = 0$, corresponding to the unique steady state. Its right eigenvector $\left|\rho_0^{\mathrm{R}}\right\rangle$ is the vectorization of the steady state density matrix $\hat{\rho}_0^{\mathrm{R}}$, while its left eigenvector $\left\langle \rho_0^{\mathrm{L}}\right|$ is Hermitian conjugate of the vectorization of the identity matrix $\hat{I}$. The remaining eigenvectors (with $E_m \neq 0$) do not represent valid density matrices, since they are typically traceless. However, this is a

natural feature of Lindblad dynamics: any physical density matrix is a linear combination of the steady state (with coefficient one to preserve trace, i.e. $a_0 \equiv 1$ in the following expansion) and traceless excited modes. This decomposition is consistent with trace preservation.

We expand the evolving state $|\rho(\lambda(t))\rangle$ with the instantaneous eigenvectors $\left|\rho_m^R(\lambda(t))\right\rangle$ as

$$|\rho(\lambda(t))\rangle = \sum_m a_m(\lambda(t)) e^{-i\theta_m(\lambda(t))} \left|\rho_m^R(\lambda(t))\right\rangle, \tag{10}$$

where $a_m$ is a time-dependent coefficient to be determined later on, and $\theta_m$ contains the dynamical phase and the Berry phase (neither necessarily to be real),

$$\theta_m(\lambda) = \int_{\lambda_i}^{\lambda} d\lambda' \left[\dot{\lambda}'^{-1} E_m(\lambda') - \varepsilon_m(\lambda')\right], \tag{11}$$

$$\varepsilon_m(\lambda') = \left\langle \rho_m^L(\lambda') | i\partial_{\lambda'} | \rho_m^R(\lambda')\right\rangle. \tag{12}$$

By substituting Eq.10 to Eq.7 and taking inner product with $\left\langle \rho_\alpha^L(\lambda)\right|$, we obtain the evolution of coefficients,

$$i\dot{a}_\alpha(t) = -\dot{\lambda} \sum_{m \neq \alpha} \left\langle \rho_\alpha^L(\lambda) \right| i\partial_\lambda \left| \rho_m^R(\lambda)\right\rangle e^{-i(\theta_m - \theta_\alpha)} a_m(t). \tag{13}$$

Since the ramping velocity $\dot{\lambda}$ is assumed to be small,[1] we treat it as a perturbative parameter. This allows us to solve the evolution equation for $a_\alpha(t)$ using a perturbative expansion in powers of $\dot{\lambda}$,

$$a_\alpha(t) = a_\alpha^{(0)}(t) + \dot{\lambda} a_\alpha^{(1)}(t) + \cdots. \tag{14}$$

In the following derivation, we keep track of at most the first order. At zeroth order, the evolution equation becomes

$$i\dot{a}_\alpha^{(0)}(t) = 0, \tag{15}$$

which implies that the zeroth-order coefficients remain constant, $a_\alpha^{(0)}(t) \equiv a_\alpha^{(0)}(t_i)$.

Even though this equation resembles the adiabatic theorem in closed systems, its physical interpretation is fundamentally different. In closed systems, adiabaticity guarantees that the occupations of eigenstates remain unchanged, due to the preservation of amplitudes up to a unit-modulus phase factor. However, in open systems, the non-steady eigenmodes of the Lindbladian have eigenvalues with negative imaginary parts. As a result, the dynamical phase in Eq.11 leads to exponential decay of all non-steady components:

$$\lim_{\dot{\lambda} \to 0} e^{-i \int d\lambda' \dot{\lambda}'^{-1} E_{m \neq 0}(\lambda')} = 0. \tag{16}$$

Therefore, in this adiabatic limit, all excitations decay away, and the final state must converge to the instantaneous steady state, regardless of the initial state. It is consistent with intuition, since the ramping evolution takes infinitely long, any population initially in non-steady modes vanishes due to their inherent decay at the zeroth order.

---

[1]According to Ref. [21] a sufficient condition for adiabatic evolution in open quantum systems (assuming non-degeneracy) is $\dot{\lambda} \ll \min(|E_{mn}|)^2$, where $E_{mn} = E_m - E_n$ is the gap between two arbitrary eigenvalues. This condition ensures that transitions between different eigenmodes are suppressed. However, if we are interested only in the expectation values of observables at the end of the ramping process, this condition can be relaxed. In particular, it suffices to require $\dot{\lambda} \ll \Gamma^2$, where $\Gamma$ is the spectral gap between the zero eigenvalue (corresponding to the steady state) and the rest of the spectrum. This is because non-steady eigenmodes are only weakly populated during slow driving, and transitions between them contribute only at higher order. As a result, their degeneracy does not affect the zeroth result and the first-order correction. Our analysis near exceptional points, presented in the Appendix A, further supports this argument — showing that degeneracies among non-steady modes do not affect the linear correction to observables.

Beyond the zeroth order, the first-order correction satisfies

$$i\dot{a}_\alpha^{(1)}(t) = -\sum_{m\neq\alpha} \langle \rho_\alpha^L | i\partial_\lambda | \rho_m^R \rangle e^{i(\theta_\alpha - \theta_m)} a_m^{(0)}(t), \tag{17}$$

which shows that the evolution of $a_\alpha^{(1)}(t)$ is coupled to other eigenstates. Integrating Eq.17 yields:

$$a_\alpha^{(1)}(t) - a_\alpha^{(1)}(t_i) = \sum_{m\neq\alpha} \left\{ i\int_{\lambda_i}^{\lambda} d\lambda'\, \dot{\lambda}'^{-1} e^{i[\theta_\alpha - \theta_m]} \langle \rho_\alpha^L | i\partial_{\lambda'} | \rho_m^R \rangle \right\} a_m^{(0)}(t_i). \tag{18}$$

Applying integration by parts, we approximate the complicated intergral to

$$i\int_{\lambda_i}^{\lambda} d\lambda'\, \dot{\lambda}'^{-1} e^{i[\theta_\alpha - \theta_m]} \langle \rho_\alpha^L | i\partial_{\lambda'} | \rho_m^R \rangle = \int_{\lambda_i}^{\lambda} \frac{d e^{i\theta_\alpha - i\theta_m}}{E_\alpha - E_m - \dot{\lambda}'(\varepsilon_\alpha - \varepsilon_m)} \frac{\langle \rho_\alpha^L | i\partial_{\lambda'} | \rho_m^R \rangle}{} $$
$$\approx \left. \frac{\langle \rho_\alpha^L(\lambda') | i\partial_{\lambda'} | \rho_m^R(\lambda') \rangle e^{i\theta_\alpha(\lambda') - i\theta_m(\lambda')}}{E_\alpha(\lambda') - E_m(\lambda')} \right|_{\lambda_i}^{\lambda}, \tag{19}$$

where in the last step we retained only the boundary term but neglected the remaining integral term and dropped the $\dot{\lambda}$-dependent correction in the denominator, as these contribute only at higher order in $\dot{\lambda}$ [19]. More discussion about the neglected terms are given in Appendix B. Therefore, the coefficient at the first order is

$$a_\alpha^{(1)}(t) = a_\alpha^{(1)}(t_i) + \sum_{m\neq\alpha} a_m^{(0)}(t_i) W_{\alpha;m}(\lambda) e^{i\theta_\alpha - i\theta_m}, \tag{20}$$

where

$$W_{\alpha;m}(\lambda) = \frac{\langle \rho_\alpha^L(\lambda) | i\partial_\lambda | \rho_m^R(\lambda) \rangle}{E_\alpha(\lambda) - E_m(\lambda)}$$
$$= -i\frac{\langle \rho_\alpha^L(\lambda) | \partial_\lambda \hat{\mathcal{L}} | \rho_m^R(\lambda) \rangle}{[E_\alpha(\lambda) - E_m(\lambda)]^2}, \tag{21}$$

where we utilize the Hellmann-Feynman theorem in open quantum systems,

$$\langle \rho_\alpha^L(\lambda) | \partial_\lambda \hat{\mathcal{L}} | \rho_m^R(\lambda) \rangle = (E_m - E_\alpha) \langle \rho_\alpha^L(\lambda) | \partial_\lambda | \rho_m^R(\lambda) \rangle .$$

Let us simplify Eq.20 further. First, for the coefficient of the steady state ($\alpha = 0$), by trace normalization of the density matrix $a_0 \equiv 1$, it is easy to see $a_0^{(1)}(t) \equiv 0$. Second, for the coefficient of non-steady states ($\alpha \neq 0$), the sum over $m \neq 0$ vanishes since the exponential factors of $e^{-i\theta_{m\neq 0}}$ all decay to zero in the adiabatic limit. Although here is a divergent factor $e^{+i\theta_\alpha}$, it is exactly canceled by its counterpart in Eq.10. Therefore, the only non-vanishing first-order contribution comes from the term with $\alpha \neq 0$ but $m = 0$, i.e., the excitation from the steady state. Importantly, we do not need to assume whether $a_{m\neq 0}(t_i)$ vanishes, since such terms do not affect the first-order correction.

Combining all of the above, we find that the state at the final time $t_f$ is given by:

$$|\rho(t_f)\rangle = e^{-i\theta_0(\lambda_f)} \times \left[ |\rho_0^R(t_f)\rangle + v \sum_{m\neq 0} W_{m;0}(\lambda_f) |\rho_m^R(t_f)\rangle \right] + \cdots. \tag{22}$$

The global prefactor $e^{-i\theta_0(\lambda_f)}$ corresponds to the Berry phase accumulated by the steady state. However, this phase is physically trivial, since the density matrix must remain Hermitian and

trace-normalized. Consequently, $e^{-i\theta_0(\lambda_f)} \equiv 1$ as shown in Ref. [31]. The proof follows directly from Eq.11 and Eq.12, noting that $E_0 \equiv 0$ and $\hat{\rho}_0^L \equiv \hat{I}$.

Switching back from the vectorized to the matrix formalism, we obtain the final expression for the density matrix:

$$\hat{\rho}(t_f) = \hat{\rho}_0^R(t_f) + v \sum_{m \neq 0} W_{m;0}(\lambda_f) \hat{\rho}_m^R(t_f) + \cdots . \tag{23}$$

This result clearly shows that the first-order correction is memoryless: it depends only on the final value of the control parameter and its velocity, but is independent of the initial state or the details of the ramping path. Therefore, as long as the final parameters (and their derivatives) are the same, the resulting state after ramping will be identical up to first order. In Appendix B, we demonstrate that the second-order correction has path-dependent contributions.

## 2.3 First-order correction of observable response

In this section, we specify the location of the control parameter $\lambda$ within the Lindbladian and compute the expectation value of an observable after the ramping process.

### 2.3.1 Ramping Hamiltonian

We first consider ramping a coherent parameter in the Hamiltonian, such that

$$\hat{H}(\lambda(t)) = \hat{H}_0 + \lambda(t)\hat{V} , \tag{24}$$

where $\hat{V}$ is a Hermitian operator associated with the ramped term. In the doubled space, the corresponding derivative of the Lindbladian matrix with respect to $\lambda$ is $\partial_\lambda \hat{\mathcal{L}} = \hat{I} \otimes \hat{V} - \hat{V}^T \otimes \hat{I}$.

Now, suppose we measure an observable $\hat{O}$ at the end of the ramping. The deviation of its expectation value from the steady-state value is given by

$$\Delta O = -iv \sum_{m \neq 0} \frac{\text{Tr}(\hat{O}\hat{\rho}_m^R)}{E_m^2} \text{Tr}(\hat{\rho}_m^{L\dagger}[\hat{V}, \hat{\rho}_0^R]) + \cdots , \tag{25}$$

where $E_m$ and $\hat{\rho}_{0,m}^{L,R}$ are all evaluated at $\lambda_f$.

The first-order correction coefficient in Eq.25 can be related to the retarded Green's function of the instantaneous steady state governed by the Lindbladian at $\lambda_f$. In the time domain, the retarded Green's function $G^R(t)$ is defined as $G^R(t) = -i\Theta(t)\langle[\hat{O}(t), \hat{V}(0)]\rangle$ [57]. By moving from the Heisenberg picture to the Schrödinger picture and applying the eigenmode decomposition of the Lindbladian (see Eq.9), the retarded Green's function can be written as:

$$\begin{aligned}
G^R(t) &= -i\Theta(t)\text{Tr}\{\hat{O}(t)[\hat{V}(0), \hat{\rho}_0^R]\} \\
&= -i\Theta(t)\text{Tr}\{\hat{O}e^{-i\hat{\mathcal{L}}t}([\hat{V}, \hat{\rho}_0^R])\} \\
&= -i\Theta(t) \sum_{m \neq 0} \text{Tr}(\hat{O}\hat{\rho}_m^R) \text{Tr}(\hat{\rho}_m^{L\dagger}[\hat{V}, \hat{\rho}_0^R]) e^{-iE_m t} .
\end{aligned} \tag{26}$$

Note that the $m = 0$ term drops out of the sum, since the left eigenvector $\hat{\rho}_0^L = \hat{I}$ and trace of a commutator is zero.

Transforming this expression to the frequency domain via Fourier transform yields the Lehmann representation of the retarded Green's function for open quantum systems:

$$G^R(\omega) = \sum_{m \neq 0} \frac{\text{Tr}(\hat{O}\hat{\rho}_m^R) \text{Tr}(\hat{\rho}_m^{L\dagger}[\hat{V}, \hat{\rho}_0^R])}{\omega - E_m} . \tag{27}$$

Taking the first derivative with respect to frequency at zero frequency, we obtain:

$$\frac{\partial G^{\mathrm{R}}}{\partial \omega}\bigg|_{\omega=0} = \sum_{m\neq 0} \frac{-1}{E_m^2}\mathrm{Tr}\left(\hat{O}\hat{\rho}_m^{\mathrm{R}}\right)\mathrm{Tr}\left(\hat{\rho}_m^{\mathrm{L}\dagger}\left[\hat{V},\hat{\rho}_0^{\mathrm{R}}\right]\right),\tag{28}$$

which exactly matches the first-order correction coefficient in Eq.25 except for an imaginary unit overall factor. Thus, we obtain the concise relation:

$$\Delta O = \mathrm{i}\frac{\partial G^{\mathrm{R}}(\omega)}{\partial \omega}\bigg|_{\omega=0}\cdot v + \cdots.\tag{29}$$

This result recovers the known conclusion for closed quantum systems [19], now extended to the open-system setting.

*Discussion*. Let us discuss several situations where the first-order coefficient may vanish. In closed systems, when $\hat{O} = \hat{V}$, the retarded Green's function is an even function of frequency due to the real energy spectrum and Hermitian dynamics. This can be seen from its Lehmann representation:

$$G^{\mathrm{R}}(\omega) = \sum_{m\neq 0}\left[\frac{|O_{0m}|^2}{\omega-(E_m-E_0)} - \frac{|O_{0m}|^2}{\omega+(E_m-E_0)}\right],\tag{30}$$

where $O_{0m} = \langle\phi_0|\hat{O}|\phi_m\rangle$ with $\phi_m$ Hamiltonian eigenstates and $m = 0$ the ground state. The evenness of $G^{\mathrm{R}}(\omega)$ implies that its derivative at zero frequency vanishes, resulting in no first-order correction. However, this cancellation generally fails in open systems due to the complex spectrum of non-Hermitian dynamics. For simplicity, we illustrate this using a non-Hermitian Hamiltonian rather than the full Lindbladian. In such a case, the eigenenergies acquire imaginary parts, and the Green's function becomes $G^{\mathrm{R}}(\omega) = \sum_{m\neq 0}\left[\frac{|O_{0m}|^2}{\omega-(E_m-E_0)} - \frac{|O_{0m}|^2}{\omega+(E_m-E_0)^*}\right]$. The complex conjugate in the second term ensures that poles lie in the lower half-plane for dynamical stability and causality, breaking the symmetry under $\omega \to -\omega$. Because of this asymmetry, the derivative at zero frequency does not cancel in general. We will demonstrate an explicit example in the 3.1 section.

There is another symmetry-based condition that can cause the first-order correction to vanish. In closed systems, this occurs when the Hamiltonian possesses an anti-unitary symmetry such as time-reversal. While time-reversal symmetry is generally not preserved in Lindbladian dynamics, charge-conjugation symmetry is not forbidden. Suppose there exists an anti-unitary operator $\hat{\mathbf{C}}$ such that

$$\hat{\mathbf{C}}\hat{\mathcal{L}}[\hat{\bullet}]\hat{\mathbf{C}}^{-1} = -\hat{\mathcal{L}}[\hat{\mathbf{C}}\hat{\bullet}\hat{\mathbf{C}}^{-1}], \qquad \hat{\mathbf{C}}\hat{\rho}_0^{\mathrm{R}}\hat{\mathbf{C}}^{-1} = \hat{\rho}_0^{\mathrm{R}},\tag{31}$$

$$\hat{\mathbf{C}}\hat{V}\hat{\mathbf{C}}^{-1} = \pm\hat{V}, \qquad\qquad \hat{\mathbf{C}}\hat{O}\hat{\mathbf{C}}^{-1} = \pm\hat{O},\tag{32}$$

then the first-order correction vanishes (a proof is provided in the Appendix C).

Finally, there exists a trivial condition that makes the first-order correction vanish. It is an infinite-temperature steady state $\hat{\rho}_0^{\mathrm{R}} = \hat{I}$, which could be achieved by Hermitian jump operators.

### 2.3.2 Ramping dissipation

In open quantum systems, it is also possible to change the dissipation. Suppose that we are ramping one of the dissipation strengths, $\lambda(t) = \kappa_i(t)$. In the doubled space, the derivative of the Lindbladian is:

$$\partial_\lambda\hat{\mathcal{L}} = \mathrm{i}\left[2\hat{J}_i^* \otimes \hat{J}_i - \hat{I}\otimes\hat{J}_i^\dagger\hat{J}_i - \left(\hat{J}_i^\dagger\hat{J}_i\right)^{\mathrm{T}}\otimes\hat{I}\right].\tag{33}$$

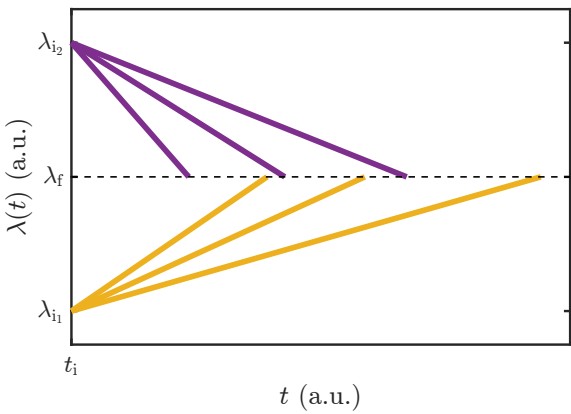

Figure 1: Illustration of our linear ramping protocol for a control parameter $\lambda(t)$. Two sets of ramping functions (in different colors) are shown, each with distinct initial values $\lambda_i$ but a shared final value $\lambda_f$. For a given initial value, we may start either from the corresponding instantaneous steady state or from a non-steady state. Within each set (i.e., for each color), we compute the post-ramping observable expectation values as functions of the ramping velocity $v$ (slope).

At the end of ramping $t_f$, the expectation value deviation from steady state of $\hat{O}$ reads:

$$\Delta O = v \sum_{m \neq 0} \frac{1}{E_m^2} \mathrm{Tr}\left(\hat{O}\hat{\rho}_m^{\mathrm{R}}\right) \times \mathrm{Tr}\left(2\hat{\rho}_m^{\mathrm{L}\dagger}\hat{J}_i\hat{\rho}_0^{\mathrm{R}}\hat{J}_i^{\dagger} - \hat{\rho}_m^{\mathrm{L}\dagger}\left\{\hat{J}_i^{\dagger}\hat{J}_i, \hat{\rho}_0^{\mathrm{R}}\right\}\right) + \cdots . \tag{34}$$

This result is naturally related to a higher-order retarded correlation function often appearing in non-Hermitian linear response theory [38, 41]. We define the high-order (retarded) correlation function as

$$\mathcal{C}^{\mathrm{R}}(t) = -\Theta(t)\left\langle 2\hat{J}_i^{\dagger}(0)\hat{O}(t)\hat{J}_i(0) - \left\{\hat{O}(t), \hat{J}_i^{\dagger}(0)\hat{J}_i(0)\right\}\right\rangle . \tag{35}$$

Using the cyclic property of the trace (as in Eq.26), its Fourier transform becomes:

$$\mathcal{C}^{\mathrm{R}}(\omega) = -\sum_{m \neq 0} \frac{\mathrm{i}}{\omega - E_m} \mathrm{Tr}\left(\hat{O}\hat{\rho}_m^{\mathrm{R}}\right) \times \mathrm{Tr}\left(2\hat{\rho}_m^{\mathrm{L}\dagger}\hat{J}_i\hat{\rho}_0^{\mathrm{R}}\hat{J}_i^{\dagger} - \hat{\rho}_m^{\mathrm{L}\dagger}\left\{\hat{J}_i^{\dagger}\hat{J}_i, \hat{\rho}_0^{\mathrm{R}}\right\}\right) . \tag{36}$$

Then, the result can be shown as

$$\Delta O = \mathrm{i}\frac{\partial \mathcal{C}^{\mathrm{R}}(\omega)}{\partial \omega}\Bigg|_{\omega=0} \cdot v + \cdots . \tag{37}$$

This expression mirrors the structure of Eq. 29 but involves a higher-order correlation function arising from dissipative perturbations.

*Discussion*. The first-order correction coefficient also vanishes under certain symmetry or structural conditions. A trivial case is that if the jump operators $\hat{J}_i$ are Hermitian and the steady state is the identity matrix. Alternatively, the correction vanishes if the system possesses a charge-conjugation symmetry $\hat{\mathbf{C}}$ such that

$$\hat{\mathbf{C}}\hat{J}_i\hat{\mathbf{C}}^{-1} \propto \hat{J}_i, \qquad \hat{\mathbf{C}}\hat{O}\hat{\mathbf{C}}^{-1} = \hat{O} . \tag{38}$$

The high-order retarded correlation function $\mathcal{C}^{\mathrm{R}}(t)$ introduced here also arises in the context of non-Hermitian linear response theory. Its behavior encodes nontrivial dynamical information. For example, it can distinguish between one-body and two-body loss processes and can serve as a diagnostic for the presence of well-defined quasiparticles in the system.

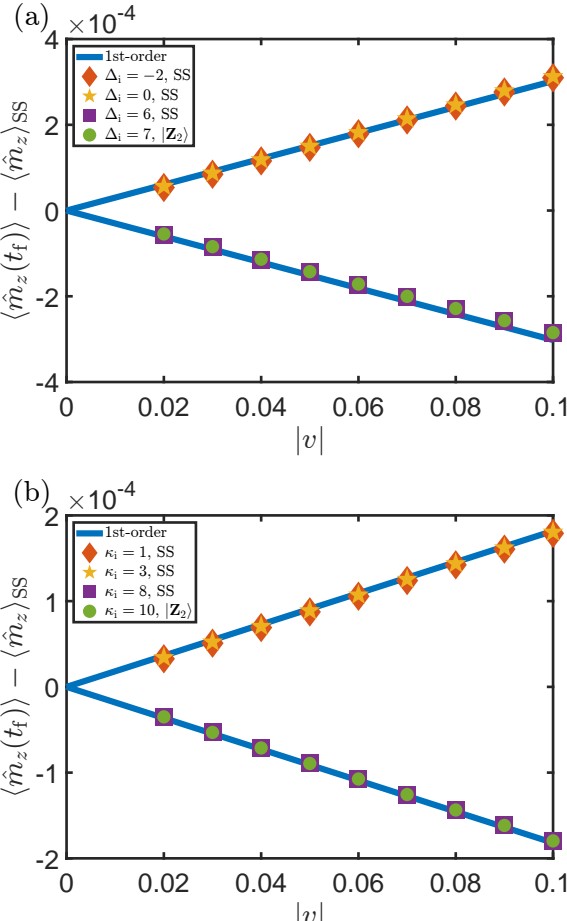

Figure 2: Deviation of the final observable expectation value $\frac{1}{N}\sum_i \langle \hat{Z}_i \rangle$ from the steady-state value in the dissipative PXP model. Numerical results from different initial states (colored markers) all converge to the analytical first-order correction (blue solid lines) in the slow-ramping limit. We set system size $N = 8$. Initial states include both instantaneous steady states (SS) and staggered $\mathbf{Z}_2$ product states. (a) Ramping the atomic detuning $\Delta(t)$ in the Hamiltonian, with final value $\Delta_{\mathrm{f}} = 3$, and fixed dissipation strength $\kappa = 1$. The horizontal axis shows the absolute value $|v|$ of the ramping velocity. Hence, the curves of initial states with $\lambda_{\mathrm{i}} > \lambda_{\mathrm{f}}$ have opposite slopes. (b) Ramping the dissipation strength $\kappa(t)$ with final value $\kappa_{\mathrm{f}} = 6$, keeping $\Delta = 0$.

## 3 Numerical verification

In this section, we numerically test our analytical predictions in several many-body open quantum systems. We employ linear ramping protocols, where the control parameter evolves with a constant velocity $\dot{\lambda}(t) \equiv v$ throughout the ramping process.

### 3.1 Dissipative PXP model

We first study a dissipative PXP model [58–61], describing a one-dimensional Rydberg atom chain with blockade radius equal to the lattice spacing. The Hamiltonian is given by

$$\hat{H}_{\mathrm{PXP}} = \sum_i \hat{P}_{i-1}\hat{X}_i\hat{P}_{i+1} + \Delta \hat{Z}_i \,, \tag{39}$$

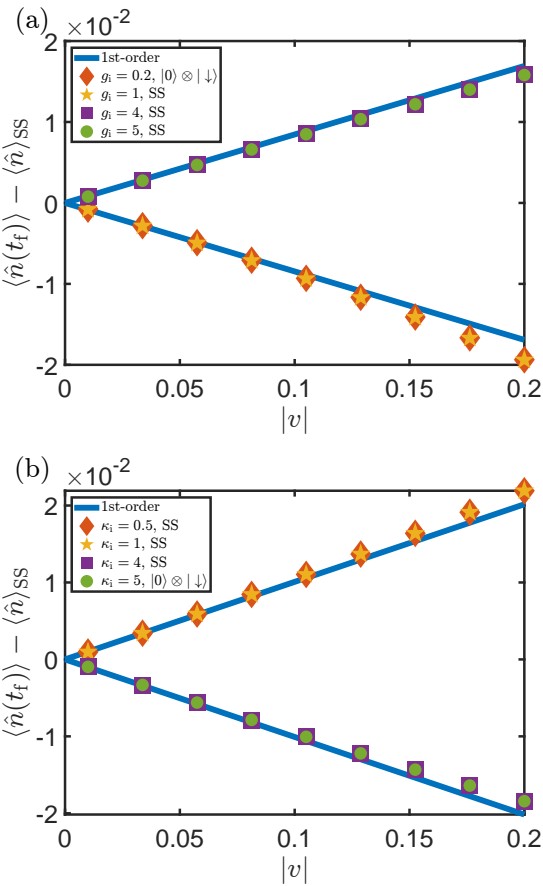

Figure 3: Deviation of the final observable $\langle \hat{n} \rangle$ from its steady-state value in the dissipative Dicke model. Results from different initial states (colored markers) agree well with the analytical first-order correction (blue solid lines) in the slow-ramping limit. We set parameters $\omega_a = 1$, $\omega_z = 2$, and $N = 4$. Initial states include instantaneous steady states (SS) as well as trivial vacuum states $|0\rangle \otimes |\downarrow\rangle$. (a) Ramping the atom-photon interaction strength $g(t)$, with final value $g_f = 3$, and fixed cavity decay rate $\kappa = 3$. The horizontal axis shows the absolute value $|v|$ of the ramping velocity; trajectories with $\lambda_i > \lambda_f$ exhibit reversed slopes. (b) Ramping the dissipation strength $\kappa(t)$ to a final value $\kappa_f = 3$, while keeping $g = 2$ fixed.

where $\hat{P}_i = |g\rangle_i \langle g|_i$ is the projection operator to the ground state, and $\hat{X}_i$ and $\hat{Z}_i$ represent the corresponding Pauli matrices, $\Delta$ is the detuning. To account for spontaneous emission from the Rydberg state, we include Lindblad dissipation with uniform decay rate $\kappa$. The full master equation reads

$$i\partial_t \hat{\rho} = \left[\hat{H}_{\text{PXP}}, \hat{\rho}\right] + i\kappa \sum_i \left(2\hat{\sigma}_i^- \hat{\rho} \hat{\sigma}_i^+ - \left\{\hat{\sigma}_i^+ \hat{\sigma}_i^-, \hat{\rho}\right\}\right), \tag{40}$$

where $\hat{\sigma}_i^\pm = 1/2 \left(\hat{X}_i \pm i\hat{Y}_i\right)$ are the spin raising and lowering operators. We perform our simulations on a chain of $N = 8$ atoms with periodic boundary conditions.

To start with, we ramp the detuning parameter $\Delta(t)$. We measure the bulk magnetization $\hat{m}_z = \sum_i \hat{Z}_i / N$. This corresponds to the special case $\hat{O} = \hat{V}$ discussed earlier, where the linear correction vanishes in closed systems, but can be finite in open systems. The results of Lindbladian evolution and the first-order correction from Eq.29 are shown in Fig.2 (a). We choose two types of initial states: the instantaneous steady state (SS), and a staggered non-steady state $|\mathbf{Z}_2\rangle = |\uparrow\downarrow\uparrow\downarrow\uparrow\downarrow\uparrow\downarrow\rangle$. In all cases, our analytical prediction matches the numerical results very well.

Next, we consider ramping the dissipation strength $\kappa(t)$. As shown in Fig.2(b), the first-order correction again agrees closely with numerical results, confirming the applicability of our response theory to both Hamiltonian and dissipative ramping protocols.

### 3.2 Dissipative Dicke model

The second model we pick is the dissipative Dicke model [62–67]. It describes a single-mode cavity coupled to a collective spin. The Hamiltonian is given by

$$\hat{H}_{\mathrm{DM}} = \omega_a \hat{a}^\dagger \hat{a} + \omega_z \hat{S}_z + \frac{2g}{\sqrt{2S}} \hat{S}_x \left(\hat{a}^\dagger + \hat{a}\right), \tag{41}$$

where $\hat{a}$ stands for the cavity mode with frequency $\omega_a$, $\hat{S}_{x,z}$ are the collective spin operators with totoal spin $S$, $\omega_z$ represents the effective Zeeman magnetic field, $g$ is the atom-photon interaction strength. Since the photons are leaking out of the cavity, the Lindblad equation reads

$$\mathrm{i}\partial_t \hat{\rho} = \left[\hat{H}_{\mathrm{DM}}, \hat{\rho}\right] + \mathrm{i}\kappa \left(2\hat{a}\hat{\rho}\hat{a}^\dagger - \left\{\hat{a}^\dagger \hat{a}, \hat{\rho}\right\}\right), \tag{42}$$

where $\kappa$ is the cavity line width. In our simulation, we use $S = 2$ and photon cutoff of 6.

First, we choose to ramp $g(t)$ and measure the post-raming photon occupation $\langle \hat{n} \rangle$ [Fig.3 (a)]. As initial states, we use either the instantaneous steady state (SS) or a trivial product state $|0\rangle \otimes |S_z = -S\rangle$ consisting of the photon vacuum and a fully spin-polarized state. In all cases, our numerical results agree well with the analytical first-order correction.

Next, we ramp the dissipation strength $\kappa(t)$. The results, shown in Fig. 3(b), again confirm the validity of our linear response correction in the slow-ramping limit.

## 4 Conclusion

In summary, we generalized the non-adiabatic linear response theory. By vectorization mapping, the evolution of the density matrix under Lindbladian resembles that of wave function under Hamiltonian. In the limit of slow ramping, we expand the coefficients to the first order of the ramping velocity. We find the first-order correction is memoryless, that is to say, only depends on the final parameters and their derivatives. Then, we explore the observable expectation value at the end of ramping. Specifically, if the ramping occurs in the Hamiltonian, the first-order correction of observable expectation value is similar to in the closed systems, which is the derivative of the retarded Green's function. While we ramp the dissipation, it is related to a higher-order correlation function. By numerically solving the evolution of two dissipative many-body models as examples, we find this non-adiabatic linear response theory works well in the slow ramping regime. Overall, our work provides a framework for probing many-body correlations in open quantum systems through controlled ramping, offering a valuable tool for quantum simulation experiments.

## Acknowledgments

**Funding information** This work is supported by NSFC (Grant No. GG2030007011 and No. GG2030040453) and Innovation Program for Quantum Science and Technology (No.2021ZD0302004).

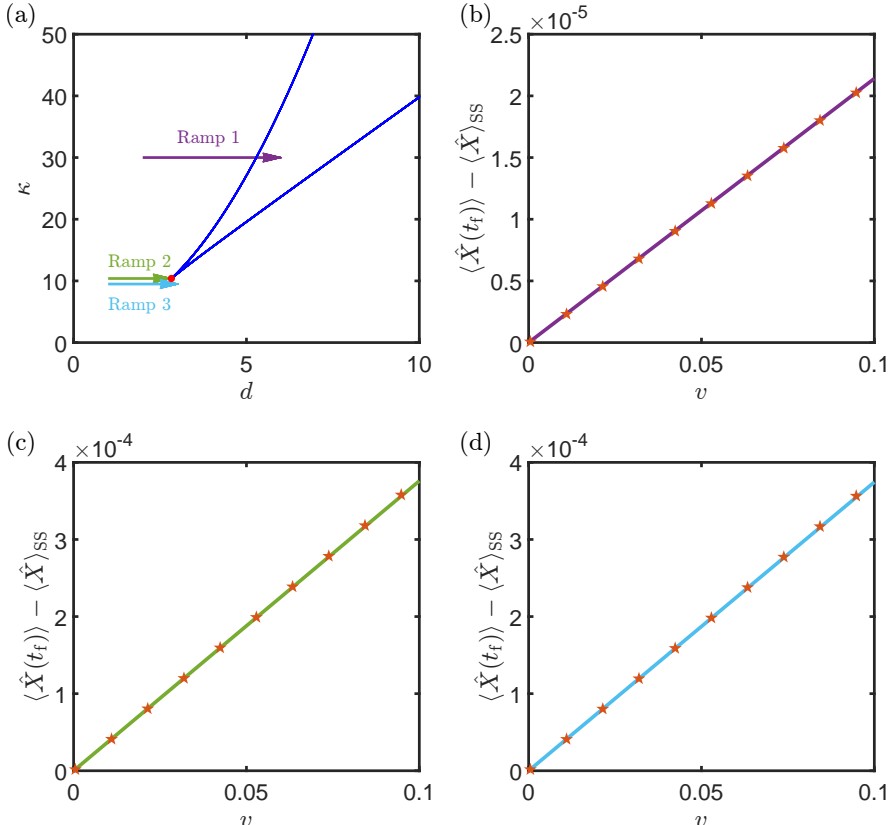

Figure 4: (a) By calculating the Lindbladian spectrum of the dissipative qubit model, We have shown two curves of 2nd-order EPs (blue lines) ending up at a 3rd-order EP (red point). We simulate the ramping in 3 paths shown in colored arrows. The shift between Ramp 2 and 3 is exaggerated for readability. (b) Ramp 1: through an EP. The result about post-ramping observable deviation of $\langle \hat{X} \rangle$ from the steady state versus the ramping velocity. Red stars represent results from Lindblad evolution. The corresponding color solid line is our first-order correction. (c) Ramp 2: end at an EP. (d) Ramp 3: end near an EP.

# A Discussion about Lindbladian exceptional points during the ramping

In the main text, we assume that the Lindbladian would not encounter any exceptional points (EPs) along the ramping path. However, even whether there exist any EPs and where they are in a many-body spectrum are inherently quite challenging questions to answer. Granted that we are able to locate where they are, then will these EPs invalidate our conclusion? Fortunately, by studying a tractable model, we can show that EPs will not make any difference.

We investigate a dissipative qubit model with spontaneous decay [56]. This is an appropriate model since the dimension of doubled space is 4, analytical solutions are not excluded by Abel-Ruffini theorem. We set the Hamiltonian is $\hat{H} = \hat{\sigma}_z + d \cdot \hat{\sigma}_x$. Considering spontaneous decay $\hat{J} = \hat{\sigma}^-$ with strength $\kappa$, the Lindbladian matrix reads

$$\hat{\mathcal{L}} = \begin{pmatrix} -2\mathrm{i}\kappa & d & -d & 0 \\ d & -2-\mathrm{i}\kappa & 0 & -d \\ -d & 0 & 2-\mathrm{i}\kappa & d \\ 2\mathrm{i}\kappa & -d & d & 0 \end{pmatrix}. \tag{A.1}$$

By solving the eigensystem, we draw the exceptional structures on Fig.4. There are two curves of second-order EPs ending up at a third-order EP at $d = 2\sqrt{2}, \kappa = 6\sqrt{3}$. There is a no-go theorem that the coalescence cannot happen on the null eigenvalue (steady state), because the trace of nonsteady eigenstates cannot change from 0 to 1 suddenly [68].

We test our results in 3 ways: 1. ramping through an EP; 2. ramping ends at an EP; 3. ramping ends near an EP.

1. We ramp $d(t)$ from $d_i = 2$ to $d_f = 6$ while keeping $\kappa = 30$. We find that our first-order correction is consistent with Lindblad evolution.

2. We ramp $d(t)$ from $d_i = 1$ to $d_f = \sqrt{2}$ while keeping $\kappa = 6\sqrt{3}$. This is the case when the 3 non-zero eigenvalues and their eigenvectors coalesce. So our derivation is wrong at the very beginning (Eq.10), and the expansion is not complete. Consequently, Eq.23 and Eq.25 are invalid. Actually, they become ill-defined at EP, since the coalesced left eigenvectors are orthogonal to their corresponding right eigenvectors at the EP (also called self-orthogonal). So we cannot find a proper and reasonable way to normalize these eigenvectors. However, the retarded Green's function should have its physical meaning and must be well-defined. Although as a result of EP, the retarded Green's function cannot be expressed as a naïve Lehmann spectral representation in Eq.27. Then we test the Eq.29 by directly calculating $G^R$ by definition. It indeed gives the correct correction.

3. We ramp $d(t)$ from $d_i = 1$ to $d_f = \sqrt{2} + 0.01$ while keeping $\kappa = 6\sqrt{3}$. Now, Eq.27 recovers the correct Green's function. Both corrections from Eq.25 and Eq.29 match each other. And we find that they are consistent with Lindblad evolution.

*Discussion*. We have indicated that the result given by Eq.29 is continuous and always correct. It is because the discontinuity of EP manifests itself on the sudden reduction in the dimension of the eigenspace. However, the evolution should be continuous. It can be calculated without performing diagonalization. We can imagine that the ramping evolution is perturbed to circumvent those EPs. This is always achievable, since the set of all EPs has zero measure in the parameter space. Explicitly, one way to circumvent an EP during the ramping is to skip it with a small step,

$$\mathcal{T}e^{-i\int_{t_i}^{t_f} dt \hat{\hat{\mathcal{L}}}(t)} = \lim_{\epsilon \to 0} \mathcal{T}e^{-i\int_{EP+\epsilon}^{t_f} dt \hat{\hat{\mathcal{L}}}(t)} e^{-i\int_{t_i}^{EP-\epsilon} dt \hat{\hat{\mathcal{L}}}(t)} . \tag{A.2}$$

All of our derivation is then well-defined and correct for the perturbed evolution. So the final observable expectation value or the Green's function will not be influenced by EPs, and the physics is not sensitive to EPs. In practice, experimental noise may shift the exact position of EPs but the continuity claims will still hold.

## B Validity of the linear approximation

In the main text, the linear response of the observable with respect to the ramping velocity $v$ was derived under the assumption of slow driving. To better understand the regime of validity of this approximation and the nature of higher-order corrections, it is helpful to consider a formal Taylor expansion of the observable expectation value at the end of the ramping process:

$$\langle \hat{O}(t_f) \rangle = \langle \hat{O} \rangle_{NESS} + \alpha \dot{\lambda} + \beta \dot{\lambda}^2 + \cdots . \tag{B.1}$$

The linear regime is expected to hold when $\dot{\lambda} \ll \alpha/\beta$. Furthermore, if $\beta$ diverges, this adiabatic condition breaks down entirely, and the linear approximation becomes invalid. However, the coefficient $\beta$ of the second-order term generally depends on the ramping protocol and the initial state, and incorporates memory effects. Its behavior is not universal and must be assessed case by case. As such, it is difficult to evaluate in a universal analytic form.

To illustrate this point, one can explicitly identify the two terms that are neglected in Eq. (16), both of which contribute to the second-order correction in the ramping velocity $\dot{\lambda}$. Although these terms are of order $\mathcal{O}(\dot{\lambda})$, they appear in the first-order expansion coefficient and thus contribute to the second-order correction in the observable.

The first neglected term originates from the correction to the denominator in the boundary terms. It gives rise to a path-independent boundary correction:

$$\dot{\lambda} \frac{[\varepsilon_\alpha(\lambda') - \varepsilon_m(\lambda')] e^{i[\theta_\alpha(\lambda') - \theta_m(\lambda')]} \left\langle \rho_\alpha^L(\lambda') \middle| i\partial_{\lambda'} \middle| \rho_m^R(\lambda') \right\rangle}{[E_\alpha(\lambda') - E_m(\lambda')]^2} \Bigg|_{\lambda_i}^{\lambda}. \tag{B.2}$$

The second neglected term is the remainder from the integration by parts:

$$\int_{\lambda_i}^{\lambda} d\lambda' \, e^{i[\theta_\alpha(\lambda') - \theta_m(\lambda')]} \, \partial_{\lambda'} \left[ \frac{\left\langle \rho_\alpha^L(\lambda') \middle| i\partial_{\lambda'} \middle| \rho_m^R(\lambda') \right\rangle}{E_\alpha(\lambda') - E_m(\lambda')} \right]. \tag{B.3}$$

This term is path-dependent and contains an oscillatory integrand. According to the Riemann-Lebesgue lemma, integrals of smooth functions multiplied by rapidly oscillating phases tend to decay as the oscillation frequency increases. In the present context, the phase oscillates with a frequency proportional to $1/\dot{\lambda}$, so the integral scales as $\mathcal{O}(\dot{\lambda})$ in the slow-ramping limit.

Consequently, the precise boundary of the linear-response regime is protocol-dependent and should be evaluated case by case. However, in the slow-ramping limit, once a linear regime is identified, the first-order result provides the correct slope. This strategy has proven effective in closed systems and is expected to be similarly applicable in open-system dynamics, offering practical guidance for experimental implementation.

# C  Proof of vanishment of the first-order correction under charge-conjugation symmetry

Here, we give a detailed proof of why the first-order coefficient would vanish under charge-conjugation symmetry. This anti-unitary symmetry can be explicitly expressed as $\hat{\mathbf{C}} = \hat{U}\mathbf{K}$, where $\hat{U}$ is a unitary operator and $\mathbf{K}$ represents the complex conjugate. We focus on the case when

$$\hat{\mathbf{C}}\hat{\mathcal{L}}[\hat{\bullet}]\hat{\mathbf{C}}^{-1} = -\hat{\mathcal{L}}[\hat{\mathbf{C}}\hat{\bullet}\hat{\mathbf{C}}^{-1}], \qquad \hat{\mathbf{C}}\hat{\rho}_0^R\hat{\mathbf{C}}^{-1} = \hat{\rho}_0^R, \tag{C.1}$$

$$\hat{\mathbf{C}}\hat{V}\hat{\mathbf{C}}^{-1} = -\hat{V}, \qquad \hat{\mathbf{C}}\hat{O}\hat{\mathbf{C}}^{-1} = -\hat{O}. \tag{C.2}$$

Other cases of different sign or about jump operators $\hat{J}_i$ are similar to generalize.

Since the Lindbladian commutes with charge-conjugation, so all eigenstates can be classified into the following three cases:

$$\hat{\mathbf{C}}\hat{\rho}_m^R\hat{\mathbf{C}}^{-1} = \hat{\rho}_m^R, \tag{C.3}$$

$$\hat{\mathbf{C}}\hat{\rho}_m^R\hat{\mathbf{C}}^{-1} = -\hat{\rho}_m^R, \tag{C.4}$$

$$\hat{\mathbf{C}}\hat{\rho}_m^R\hat{\mathbf{C}}^{-1} = \hat{\rho}_{\bar{m}}^R = \hat{\rho}_m^{R\dagger}, \tag{C.5}$$

where the first two cases can happen when $\mathrm{Re}(E_m) = 0, \hat{\rho}_m^R = \hat{\rho}_m^{R\dagger}$, and in the last case charge conjugation would relate $\hat{\rho}_m^R$ to its Hermitian conjugate $\hat{\rho}_{\bar{m}}^R = \hat{\rho}_m^{R\dagger}$ which is also an eigenstate with eigenvalue $E_{\bar{m}} = -E_m^*$.

Let us see what will happen to the coefficient,

$$\frac{\partial G^R}{\partial \omega}\bigg|_{\omega=0} = \sum_{m \neq 0} \frac{-1}{E_m^2} \mathrm{Tr}\left(\hat{O}\hat{\rho}_m^R\right) \mathrm{Tr}\left(\hat{\rho}_m^{L\dagger}[\hat{V}, \hat{\rho}_0^R]\right). \tag{C.6}$$

For the first case, because of the Hermiticity of $\hat{\rho}_m^{\mathrm{R}}$ and $\hat{O}$,

$$
\begin{aligned}
\mathrm{Tr}\big(\hat{O}\hat{\rho}_m^{\mathrm{R}}\big) &= \mathrm{Tr}\big(\hat{O}^*\hat{\rho}_m^{\mathrm{R}*}\big) \\
&= \mathrm{Tr}\big[\hat{U}^\dagger(-\hat{O})\hat{U}\hat{U}^\dagger\hat{\rho}_m^{\mathrm{R}}\hat{U}\big] \\
&= -\mathrm{Tr}\big(\hat{O}\hat{\rho}_m^{\mathrm{R}}\big) = 0\,.
\end{aligned}
\tag{C.7}
$$

For the second case, remember $\hat{\rho}_m^{\mathrm{L}}$ is also Hermitian and acquires the same minus sign under charge conjugation, and $\mathrm{Tr}(\hat{\bullet}^\dagger) = \mathrm{Tr}(\hat{\bullet})^* = \mathrm{Tr}(\hat{\bullet}^*)$,

$$
\begin{aligned}
\mathrm{Tr}\big(\hat{\rho}_m^{\mathrm{L}\dagger}\big[\hat{V},\hat{\rho}_0^{\mathrm{R}}\big]\big) &= \mathrm{Tr}\big(\hat{\rho}_m^{\mathrm{L}}\big[\hat{V}\hat{\rho}_0^{\mathrm{R}} - \hat{\rho}_0^{\mathrm{R}}\hat{V}\big]\big) \\
&= \mathrm{Tr}\big(\hat{\rho}_m^{\mathrm{L}}\big[\hat{\rho}_0^{\mathrm{R}}\hat{V} - \hat{V}\hat{\rho}_0^{\mathrm{R}}\big]\big)^* \\
&= \mathrm{Tr}\big(\hat{\rho}_m^{\mathrm{L}*}\big[\hat{\rho}_0^{\mathrm{R}*}\hat{V}^* - \hat{V}^*\hat{\rho}_0^{\mathrm{R}*}\big]\big) \\
&= \mathrm{Tr}\big(\hat{U}^\dagger\hat{\rho}_m^{\mathrm{L}*}\hat{U}\hat{U}^\dagger\big[\hat{\rho}_0^{\mathrm{R}*}\hat{V}^* - \hat{V}^*\hat{\rho}_0^{\mathrm{R}*}\big]\hat{U}\big) \\
&= \mathrm{Tr}\big\{-\hat{\rho}_m^{\mathrm{L}}\big[-\hat{\rho}_0^{\mathrm{R}}\hat{V} + \hat{V}\hat{\rho}_0^{\mathrm{R}}\big]\big\} \\
&= \mathrm{Tr}\big(\hat{\rho}_m^{\mathrm{L}}\big[\hat{\rho}_0^{\mathrm{R}},\hat{V}\big]\big) = 0\,.
\end{aligned}
\tag{C.8}
$$

For the third case,

$$
\begin{aligned}
\mathrm{Tr}\big(\hat{O}\hat{\rho}_m^{\mathrm{R}}\big)^* &= \mathrm{Tr}\big(\hat{O}\hat{\rho}_m^{\mathrm{R}\dagger}\big) \\
&= \mathrm{Tr}\big(\hat{O}^*\hat{\rho}_m^{\mathrm{R}*}\big) \\
&= \mathrm{Tr}\big[\hat{U}^\dagger(-\hat{O})\hat{U}\hat{U}^\dagger\hat{\rho}_m^{\mathrm{R}\dagger}\hat{U}\big] \\
&= -\mathrm{Tr}\big(\hat{O}\hat{\rho}_m^{\mathrm{R}\dagger}\big) = 0\,.
\end{aligned}
\tag{C.9}
$$

Thus, the first-order coefficient must vanish.

A minimal model that can be easy to test is that $\hat{H} = \hat{\sigma}_x, \hat{J} = \hat{\sigma}^-$. In this model, charge-conjugation is $\hat{\mathbf{C}} = \hat{\sigma}_z\mathbf{K}$. And operators change their signs under $\hat{\mathbf{C}}$: $\{\hat{\sigma}_x\}$. Operators invariant under $\hat{\mathbf{C}}$: $\{\hat{\sigma}_y,\hat{\sigma}_z\}$.

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
