# Peer review of "Non-adiabatic linear response in open quantum systems"

_SciPost Physics, doi:SciPost Phys. 19, 161 (2025)_

## Round 1 · Referee Report · Anonymous (Referee 1) · 2025-11-13

Report

In this work, the authors generalize non-adiabatic linear response theory to open quantum many-body systems with a unique steady state, incorporating ramps of both Hamiltonian and dissipation parameters. In particular, they demonstrate that the leading-order response depends only on the final parameter value. The theoretical predictions are validated numerically using dissipative PXP and dissipative Dicke models. The authors further argue that Lindbladian exceptional points do not affect their conclusions.

Overall, I believe the work is of high quality. It presents a clear and versatile framework for probing many-body correlations in open systems through controlled parameter ramps. The manuscript is well written and accessible to a broad audience, with direct experimental relevance. Therefore, I am pleased to recommend its publication in SciPost Physics.

Here are some minor questions:

The authors focus on the response of observables. How about entropic quantities? For example, does a first-order contribution appear in the system’s purity?

What happens if the ramp crosses a phase transition, such as spontaneous weak symmetry breaking (SWSSB)?

Recommendation

Publish (easily meets expectations and criteria for this Journal; among top 50%)

  • validity: top
  • significance: high
  • originality: high
  • clarity: top
  • formatting: excellent
  • grammar: excellent

Author:  Xiaotian Nie  on 2025-11-25  [id 6070]

(in reply to Report 1 on 2025-11-13)
Disclosure of Generative AI use

The comment author discloses that the following generative AI tools have been used in the preparation of this comment:

We use ChatGPT 5.1 (OpenAI) for grammar correction and tone polishing.

Category:
answer to question

Response to Referee 1

We thank Referee 1 for their very positive and encouraging report, for recommending publication in SciPost Physics, and for the careful reading of our manuscript. We are grateful in particular for the two insightful questions, which help clarify the generality and limitations of our results. Below we address these points in order.

The referee writes:

The authors focus on the response of observables. How about entropic quantities? For example, does a first-order contribution appear in the system’s purity?

Our response:

We thank Referee 1 for raising this insightful question. Let us discuss the purity, which is directly related to the Rényi-2 entropy. Starting from Eq. (23) of the manuscript, we write the final state after the ramp as

$$ \rho(t_f) = \rho_0^{R} + v \sum_{m \neq 0} W_{m0}\,\rho_m^{R} + \mathcal{O}(v^2), $$
where $\rho_0^{R}$ is the right eigenstate associated with the steady state and $\rho_m^{R}$ are the remaining right eigenmodes of the Lindbladian.

Expanding the purity to first order in $v$, we obtain

$$ \mathrm{Tr}[\rho^2(t_f)] = \mathrm{Tr}[(\rho_0^{R})^2] + 2v \sum_{m \neq 0} W_{m0}\,\mathrm{Tr}[\rho_0^{R}\rho_m^{R}] + \mathcal{O}(v^2). $$
In general, for a non-Hermitian Lindbladian the right eigenstates ${\rho_m^{R}}$ do not form an orthogonal set under the Hilbert–Schmidt inner product, so the overlaps $\mathrm{Tr}[\rho_0^{R}\rho_m^{R}]$ need not vanish. Consequently, the coefficient of the linear term in $v$ is generically nonzero, and a first-order response in the purity is expected.

The referee writes:

What happens if the ramp crosses a phase transition, such as spontaneous weak symmetry breaking (SWSSB)?

Our response:

We thank Referee 1 for this fundamental question. Our derivation explicitly relies on the condition stated in Footnote [57] of the manuscript: the ramp velocity $v$ must be much smaller than the Lindbladian gap around the steady state along the ramp protocol. This separation of scales is crucial for controlling the non-adiabatic corrections and for showing that the leading-order response depends only on the final parameter value.

At a phase transition, such as one involving spontaneous weak symmetry breaking, the Lindbladian gap closes in the thermodynamic limit. In this regime, the basic assumption underlying our expansion breaks down, and one expects qualitatively different, non-perturbative behavior. Therefore, our linear-response formula in its present form is not applicable to ramps that cross such phase transitions. A proper treatment would require a separate analysis that explicitly takes into account the critical closing of the gap, which lies beyond the scope of the current work.

---

## Round 1 · Referee Report · Anonymous (Referee 2) · 2025-11-17

Report

The manuscript develops a non-adiabatic linear response framework for open quantum systems governed by Lindbladians under slow parameter ramps, covering both Hamiltonian and dissipative ramping. It derives first-order corrections to the density matrix and observable responses, discusses vectorization of the Liouvillian, and addresses continuity issues at Lindbladian exceptional points (EPs). The authors verified their main results Eq. (29), numerically on dissipative PXP and Dicke models; they also discuss the validity of the linear approximation and a symmetry condition under which first-order corrections vanish. This manuscript is well written, and the results are timely and solid. So, I would like to recommend a publication on Scipost. I also have some questions and suggestions for the authors:
1. What are the minimal Lindbladian spectral conditions required for the linear approximation to remain controlled during the ramp?
2. In the numerical tests, how close do you approach the EP?
3. Does the proposed response formula handle cases with multiple steady states?
4. Add a discussion of physical observables most sensitive to EP crossings and how experimental noise/dissipation might affect the continuity claims.
5. Define all abbreviations and symbols at first use (e.g., EP, PXP).

Recommendation

Publish (easily meets expectations and criteria for this Journal; among top 50%)

  • validity: -
  • significance: -
  • originality: -
  • clarity: -
  • formatting: -
  • grammar: -

Author:  Xiaotian Nie  on 2025-11-25  [id 6071]

(in reply to Report 2 on 2025-11-17)
Disclosure of Generative AI use

The comment author discloses that the following generative AI tools have been used in the preparation of this comment:

We use ChatGPT 5.1 (OpenAI) for grammar correction and tone polishing.

Category:
answer to question

Response to Referee 2

We thank Referee 2 for their careful reading of our manuscript and for the very positive assessment, including the recommendation for publication in SciPost Physics. We are also grateful for the clear questions and helpful suggestions. Below we address these points in order.

The referee writes:

  1. What are the minimal Lindbladian spectral conditions required for the linear approximation to remain controlled during the ramp?

Our response:

We thank Referee 2 for this fundamental question. In our paper, we provide a necessary condition in Footnote [57], which gives an upper bound on the ramp velocity in terms of the spectral gap between the zero eigenvalue (corresponding to the steady state) and the rest of the spectrum. However, we are not able to provide a simple sufficient condition that guarantees that the linear-response regime always holds. As we emphasize in Appendix B, although we cannot specify the precise boundary of validity, once a linear regime is identified in the slow-ramping limit, the first-order result provides the correct slope.

The referee writes:

  1. In the numerical tests, how close do you approach the EP?

Our response:

We thank Referee 2 for this detailed question. In Appendix A, we discuss numerical ramps that cross, end at, or approach an exceptional point (EP). Ramp 2 in Fig. 4 corresponds to a protocol whose final parameter is tuned exactly to an EP, while Ramp 3 ends very close to the EP (with a deviation of order (0.01) in units of the relevant energy scale). In both cases, the results are consistent with our non-adiabatic linear-response prediction.

The referee writes:

  1. Does the proposed response formula handle cases with multiple steady states?

Our response:

We thank Referee 2 for this important question. As mentioned in the Introduction, our analysis is explicitly aimed at the case of a unique steady state, which is precisely what allows us to arrive at a simple and universal formula for the first-order response. In the presence of multiple steady states, our derivation does not apply: the final state is then in general not memoryless, because information about the ramping history can be encoded in the steady-state subspace. This regime has been studied in detail in Ref. [31] (Victor V. Albert, Barry Bradlyn, Martin Fraas, and Liang Jiang, Phys. Rev. X 6, 041031), where multiple steady states lead to rich geometric phenomena such as holonomy. A full treatment of this scenario lies beyond the scope of our work.

The referee writes:

  1. Add a discussion of physical observables most sensitive to EP crossings and how experimental noise/dissipation might affect the continuity claims.

Our response:

We thank Referee 2 for this comment. We have added discussion in the main text like, “Since there is a no-go theorem that EP will not happen on the steady state of a Lindbladian and our paper is talking about physics near the steady state [PHYSICAL REVIEW A 98, 042118 (2018)], so the physics is not sensitive to EP” and “Experimental noise may shift the exact position of EPs but the continuity claims will still hold”.

The referee writes:

  1. Define all abbreviations and symbols at first use (e.g., EP, PXP).

Our response:

We thank Referee 2 for this useful reminder. We have gone through the manuscript to check all abbreviations and symbols and have ensured that they are now defined at their first occurrence (including EP, PXP, and other acronyms).

Anonymous on 2025-11-26  [id 6074]

(in reply to Xiaotian Nie on 2025-11-25 [id 6071])

I think the authors have addressed my questions in a scientific way. Now I would like to recommend the publication.

---

## Editorial Decision

published